# AVG-LLaVA: A Large Multimodal Model with Adaptive Visual Granularity

## Abstract

Recently, when dealing with high-resolution images, dominant large multimodal models (LMMs) usually divide them into multiple local images and one global image, which will lead to a large number of visual tokens. In this work, we introduce AVG-LLaVA, an LMM that can adaptively select the appropriate visual granularity based on the input image and instruction. This approach not only reduces the number of visual tokens and speeds up inference, but also improves the overall model performance. Specifically, we introduce the following modules based on LLaVA-NeXT: (a) a visual granularity scaler that includes multiple pooling layers to obtain visual tokens with different granularities; (b) a visual granularity router, which includes a Transformer layer, an MLP layer, and a voter layer, used to select the appropriate visual granularity based on the image and instruction. Furthermore, we propose RGLF, a novel training paradigm that aims at aligning the granularity predicted by the router with the preferences of the LMM, without the need for additional manually annotated data. Extensive experiments and analysis show that AVG-LLaVA achieves superior performance across 11 benchmarks, as well as significantly reduces the number of visual tokens and speeds up inference (e.g., an 85.3% reduction in visual tokens and a 2.53× increase in inference speed on the AI2D benchmark). [1]

## 1 Introduction

Recently, the field of artificial intelligence (AI) has witnessed a significant advancement in large multimodal models (LMMs) (OpenAI, 2023b; Zhu et al., 2023; Dai et al., 2023; Liu et al., 2023a; 2024a), marking a further step toward artificial general intelligence (AGI). Most existing LMMs follow the structure of LLaVA (Liu et al., 2023a; 2024a), which includes a vision encoder to embed images into visual tokens and a connector to map them into the word embedding space. Subsequently, these visual tokens are fed into a large language model (LLM) (Touvron et al., 2023; OpenAI, 2023a; Chiang et al., 2023) for multimodal understanding and reasoning, alongside the word embeddings.

Due to the limitations imposed by the fixed aspect ratio (e.g., 1:1) and low resolution (e.g., 336×336) used by visual encoders (e.g., CLIP-ViT (Radford et al., 2021)), earlier LMMs face challenges in processing high-resolution images with different aspect ratios. To deal with this limitation, dominant models, such as LLaVA-NeXT (Liu et al., 2024b), dynamically divide each input high-resolution image into multiple local images. These local images are encoded separately, and their tokens are then concatenated with the tokens of the original global image. This approach will lead to longer

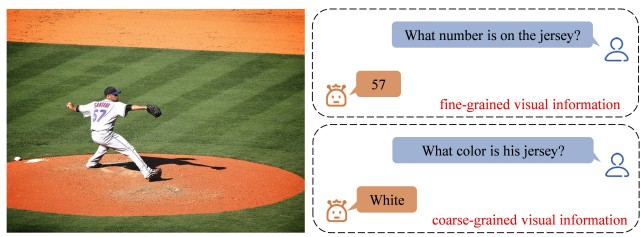

Figure 1: An example of VQA that requires information at different visual granularities. The image is from MSCOCO (Lin et al., 2014) dataset.

---

[1]We will release our source code and model upon the acceptance of our paper.

visual token sequences, such as 2880 visual tokens for a 672×672 image. However, in practice, such fine-grained visual information is not always necessary, and in some cases, coarse-grained visual information can even be more beneficial for model predictions. For instance, as shown in Figure 1, when the model is asked to recognize the number on the jersey, it requires relatively fine-grained visual information. In contrast, determining the color of the jersey only necessitates coarse-grained visual information.

In this paper, we propose *Adaptive Visual Granularity LLaVA (AVG-LLaVA)*, an LMM that can adaptively select the appropriate visual granularity based on the input image and instruction. The basic intuition behind our model is that humans only scrutinize images carefully when answering difficult questions; otherwise, a brief glance is sufficient. As displayed in Figure 2, AVG-LLaVA extends LLaVA-NeXT with a *visual granularity scaler* and a *visual granularity router*. The visual granularity scaler performs multiple rounds of pooling on visual tokens without training, each time halving the number of visual tokens, thus obtaining a series of visual features with different granularities. The visual granularity router adaptively selects the appropriate visual granularity features based on the input multi-granularity visual features and text features. By doing so, for images and instructions that do not require fine-grained details, the number of visual tokens can be reduced, which not only speeds up inference but also improves performance. This performance enhancement likely stems from the reduction of redundant information, as selecting the appropriate visual granularity makes it easier for the model to answer questions based on images effectively.

Besides, we observe that it is challenging to train the visual granularity router directly through visual instruction tuning (Liu et al., 2023a). This may be because the router cannot learn the distinctions between different visual granularities from visual instruction tuning, making it difficult to learn how to select the most appropriate visual granularity based on the image and instruction. To deal with this issue, we propose a novel training paradigm called *Ranking Granularity* to align *LMM Feedback* (*RGLF*). This paradigm aligns router probabilities of multiple granularities with LMM preferences by a ranking loss (Hadsell et al., 2006; Hopkins & May, 2011; Liu et al., 2022), effectively aiding the router in distinguishing between different visual granularities and selecting the appropriate one.

We further evaluate AVG-LLaVA on 11 benchmarks including tasks from various types (e.g., general VQA and text-oriented VQA, etc.). Extensive experimental results show that AVG-LLaVA can effectively reduce the number of visual tokens and improve inference speed (e.g., an 85.3% reduction in visual tokens and a 2.53× increase in inference speed on the AI2D benchmark) while achieving better performance under the same base LLM.

## 2 RELATED WORK

**Large Multimodal Models**  LLMs such as GPT-4 (OpenAI, 2023a), LLaMA (Touvron et al., 2023), and Gemini (Team et al., 2023) have achieved significant success in language understanding and generation. Benefiting from this, multimodal large models (LMMs) have garnered widespread attention. Flamingo (Alayrac et al., 2022), BLIP-2 (Li et al., 2023b), and LLaMA-adapter (Zhang et al., 2023) integrate a frozen visual encoder and trainable modules into a LLM, extending it into a LMMs. These models are then fine-tuned using plain image-text pairs, enabling them to process and perceive visual content. To further improve instruction-following abilities and response quality, LLaVA (Liu et al., 2023a) fine-tunes the entire model using visual instruction data generated by GPT-4. However, since these LMMs rely on CLIP-ViT to process images at a fixed resolution (e.g. 336x336), it hinders the LMMs from perceiving image details at higher resolutions.

**High-Resolution LMMs**  To perceive images with higher resolutions, Qwen-VL (Bai et al., 2023) increases the input resolution of the visual encoder to 448×448 and introduces an additional training stage. Along this line, both Vary (Wei et al., 2023) and Mini-Gemini (Li et al., 2024a) include two vision encoders: one is an additional introduced high-resolution vision encoder, and the other is the original low-resolution vision encoder. Unlike the methods mentioned above, SPHINX (Lin et al., 2023) and Monkey (Li et al., 2024b) enlarge the input image to a high resolution, and then divide it into a fixed number of local images, which are individually encoded using an image encoder to obtain local image tokens. Subsequently, the original global image tokens are concatenated with all local image tokens to feed into the LLM. Furthermore, LLaVA-NeXT (Liu et al., 2024b) enumerates various resolutions and adaptively selects the one that most closely matches the input image

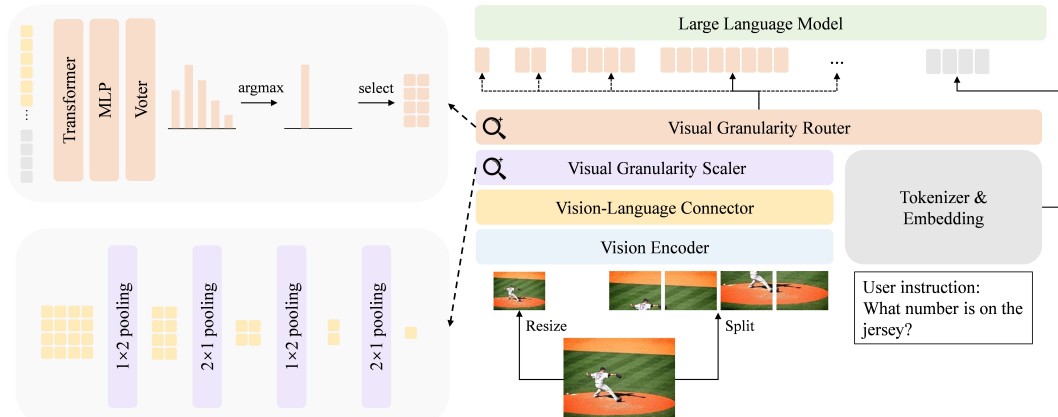

Figure 2: The architecture of AVG-LLaVA. AVG-LLaVA additionally introduces two modules based on LLaVA-NeXT: (1) Visual granularity scaler. This module consists of multiple pooling layers that progressively increase the granularity of visual features, thereby reducing the number of visual tokens; (2) Visual granularity router. This module includes a Transformer layer, an MLP layer, and a voter layer, which are used to select the appropriate granularity of visual features based on the input multi-granularity visual tokens and instruction tokens.

resolution. Although these methods can achieve better performance, they significantly increase the number of visual tokens, as the computational complexity scales quadratically with the number of input tokens, resulting in higher inference costs.

**Vision Token Reduction for LMMs**    Recently, several methods are proposed to reduce the visual tokens for LMMs, including visual token pruning and merging. For example, CrossGET (Shi et al., 2023) introduces a cross-modal token for leveraging cross-modal information to make decisions on token selection and merging. LLaVA-PruMerge (Shang et al., 2024) employs the similarity between the class token and other tokens as a key criterion for pruning and merging vision tokens. Unlike the aforementioned methods, FastV (Chen et al., 2024b) finds that most image tokens receive inefficient attention after the second decoder layer, and thus prunes half of the image tokens. Similarly, VTW (Lin et al., 2024b) adopts a more aggressive strategy to prune all visual tokens at a certain layer. Unfortunately, despite the above methods effectively reducing the number of visual tokens, they often lead to a certain degree of decline in model performance. More recently, LLaVA-$M^3$ (Cai et al., 2024) obtains multi-granularity visual features by merging visual tokens through pooling, enabling manual control of the tradeoff between inference cost and performance. Significantly different from these methods, our model can adaptively select the appropriate visual granularity based on the input image and instruction, improving model performance while reducing visual tokens.

## 3 OUR MODEL

### 3.1 MODEL ARCHITECTURE

As shown in Figure 2, in addition to the visual encoder, visual-language connector, and LLM, AVG-LLaVA introduces two additional modules on top of LLaVA-NeXT: the visual granularity scaler and the visual granularity router. The key components will be elaborated in the following.

**High-Resolution Image Encoding**    Given an input image $\mathbf{I} \in \mathbb{R}^{H \times W \times 3}$, we follow common practice (Liu et al., 2024b) to divide it into multiple smaller local images $\mathbf{I}_{local} \in \mathbb{R}^{H_v \times W_v \times 3}$, where $H_v$ and $W_v$ are the resolution that the vision encoder is originally trained for. Then, these local images are individually encoded into a $H_p \times W_p$ grid of visual tokens $\mathbf{X}_{local} \in \mathbb{R}^{H_p \times W_p \times C}$ by the image encoder, where $C$ is the dimension of the visual encoder. To preserve the global context information of the input image, we resize the original image to $(H_v, W_v)$ and encode it as global visual tokens. Finally, we map both global visual tokens and local visual tokens to the word embedding space through an MLP-based vision-language connector.

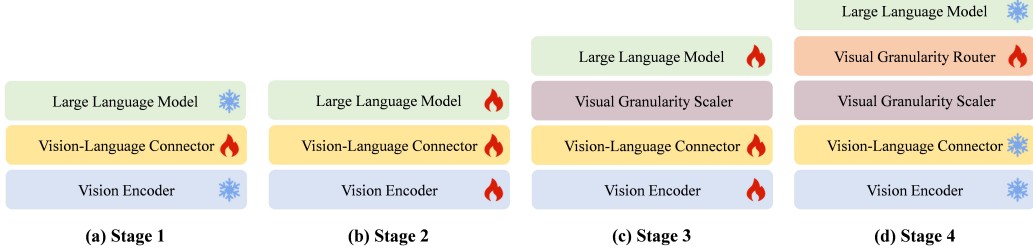

Figure 3: The overview of multi-stage training. Note that visual granularity scaler does not contain any parameters that require training.

**Visual Granularity Scaler**   This module sequentially stacks 1×2 and 2×1 average pooling layers, thereby obtaining visual features at multiple granularities and preserving the spatial information. In this work, we consider CLIP-ViT-L-336 (Radford et al., 2021) as the visual encoder, and thus each image is encoded into 24×24 grid of visual tokens. Then, these visual tokens are fed into the visual granularity scaler, obtaining visual tokens with a grid of 24×12, 12×12, 12×6 and 6×6, respectively. In this way, we can obtain visual tokens of different granularities in a fine-to-coarse manner without training.

**Visual Granularity Router**   Different visual granularity features can be considered as different experts, so the Mixture of Experts (MoE) (Shazeer et al., 2017; Komatsuzaki et al., 2023; Lin et al., 2024a) structure is particularly well-suited for selecting the appropriate visual granularity. Unlike the previous MoE studies that use linear layers as routers, we propose a multi-layer structure as illustrated in Figure 2 to select the appropriate visual granularity based on the input image and the instruction. Specifically, when dealing with an image, we first flatten and concatenate its visual tokens of all granularities to form multi-granularity visual tokens $\overline{\mathbf{X}}_v = [\mathbf{X}_v^1; \mathbf{X}_v^2; ...; \mathbf{X}_v^N]$, where $\mathbf{X}_v^i$ represents the visual tokens of the $i$-th granularity, and $N$ is the number of visual granularities[2]. Then, these visual tokens are concatenated with the filtered instruction tokens $\overline{\mathbf{X}}_{instruct}$ to serve as the input for the visual granularity router. Here, $\overline{\mathbf{X}}_{instruct}$ is obtained by calculating the cosine similarity between the original instruction tokens $\mathbf{X}_{instruct}$ and the visual tokens with original granularity $\mathbf{X}_v$, retaining the top-$k$ most relevant ones. Afterwards, we apply a single Transformer (Vaswani et al., 2017) layer to facilitate the fusion of visual tokens at different granularities with instruction tokens. Subsequently, an MLP is applied to each token to predict the appropriate visual granularity, resulting in the logits $\mathbf{Z}_{out} \in \mathbb{R}^{L \times N}$, where $L$ is the number of both visual and instruction tokens. To vote for the most appropriate visual granularity, we use a learnable weight matrix (Voter) $\mathbf{W} \in \mathbb{R}^{1 \times L}$ to aggregate the logits predicted by all tokens, yielding the final logits $\mathbf{Z}_{final} \in \mathbb{R}^{1 \times N}$. Finally, we use softmax to calculate the probability distribution of each visual granularity, where the visual tokens corresponding to the granularity with the highest probability are fed into the LLM.

### 3.2 MULTI-STAGE TRAINING

To effectively train our model, we carefully design a multi-stage training strategy, which consists of four stages, as illustrated in Figure 3.

**Stage 1: Pretraining**   During this stage, we only pretrain the vision-language connector on a plain image-caption dataset. Formally, we define the following cross-entropy loss for the next token prediction:

$$\mathcal{L}_1 = -\sum_{t=1}^{T} \log P(x_t | \mathbf{X}_v, \mathbf{X}_{c,<t}),  \tag{1}$$

---

[2]To simplify the explanation, we use a single image as an example. In practice, we include a global image and multiple local images, and each image will go through the following steps. The final result will be obtained by averaging the results of all the images.

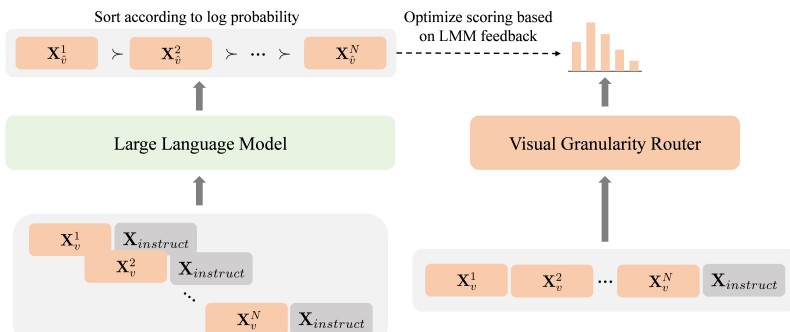

Figure 4: The overview of RGLF. Visual tokens of each granularity are concatenated with instruction tokens and then processed by the LMM to estimate the corresponding rewards. Visual granularity router optimizes the score (log probability) of each granularity based on the feedback from the LMM.

where $\mathbf{X}_{c,<t}$ are the caption tokens before the current prediction token $x_t$, and $T$ is the length of target text tokens. Note that we fix the parameters of the vision encoder and large language model, both of which have been pre-trained on large-scale data. Through this stage of training, the image features can be aligned with the pre-trained LLM word embeddings.

**Stage 2: Visual Instruction Tuning** In the second stage, we jointly train the visual encoder, vision-language connector, and LLM on high-quality visual instruction data. In this way, the LLM can be converted into an LMM, which is able to complete various multimodal tasks. Specifically, we perform next-token predictions with the following cross-entropy loss only on the answering part

$$\mathcal{L}_2 = -\sum_{t=1}^{T} \log P(x_t | \mathbf{X}_v, \mathbf{X}_{instruct}, \mathbf{X}_{a,<t}), \tag{2}$$

where $\mathbf{X}_a$ denotes the answer tokens.

**Stage 3: Multi-Granularity Visual Instruction Tuning** Following the previous stages, we introduce the visual granularity scaler. As described in Section 3.1, this module does not contain trainable parameters and thus does not need to be trained. Therefore, we use the same data as in the stage 2 to train the visual encoder, vision-language connector, and LLM, enabling them to perceive and process visual features of $N$ different granularities. Formally, the loss at this stage is formulated as

$$\mathcal{L}_3 = -\frac{1}{N} \sum_{i=1}^{N} \sum_{t=1}^{T} \log P(x_t | \mathbf{X}_v^i, \mathbf{X}_{instruct}, \mathbf{X}_{a,<t}). \tag{3}$$

**Stage 4: Ranking Granularity to Align LMM Feedback** Lastly, we introduce the visual granularity router into the model training, where all other modules are frozen, and only the router is trained. This stage allows the model to select the appropriate visual granularity based on the input image and instruction. Intuitively, a straightforward approach to training the router is to use the same visual instruction fine-tuning method as in previous stages. However, we find that the router trained with this method performs poorly. This could be due to the difficulty of visual instruction fine-tuning in effectively enabling the router to learn the differences between different visual granularities.

To address the above issue, we propose RGLF, as illustrated in Figure 4, where the router is trained with a ranking loss, utilizing the feedback from the LMM fine-tuned with multi-granularity visual instructions as the ranking criterion. Concretely, for the given image and instructions, we let the LMM predict answers using visual tokens of different granularity $\mathbf{X}_v^i$ and calculate their respective log probabilities. Then, based on these log probabilities, we sort $\mathbf{X}_v^1; \mathbf{X}_v^2; ...; \mathbf{X}_v^N$ in a descending order to obtain $\mathbf{X}_{\hat{v}}^1; \mathbf{X}_{\hat{v}}^2; ...; \mathbf{X}_{\hat{v}}^N$. Given the visual tokens $\mathbf{X}_{\hat{v}}^i$ of the $i$-th granularity, we directly consider those tokens $(\mathbf{X}_{\hat{v}}^1; \mathbf{X}_{\hat{v}}^2; ...; \mathbf{X}_{\hat{v}}^{i-1})$ ranked above it as positive examples and the remaining

tokens $(\mathbf{X}_{\hat{v}}^{i+1}; \mathbf{X}_{\hat{v}}^{i+2}; ...; \mathbf{X}_{\hat{v}}^{N})$ as negative ones. Afterwards, we use the router to give scores (log probability) $s_i$ for each $\mathbf{X}_{\hat{v}}^i$:

$$s_i = \log P(g_i|\overline{\mathbf{X}}_v, \overline{\mathbf{X}}_{instruct}), \tag{4}$$

where $g_i$ denotes the $i$-th granularity predicted by the router based on multi-granularity visual tokens $\overline{\mathbf{X}}_v$ and filtered instruction tokens $\overline{\mathbf{X}}_{instruct}$. Since we expect the router to assign higher probabilities to more appropriate visual granularities, the ranking loss is defined as follows:

$$\mathcal{L}_{rank} = \sum_{i=1} \sum_{j>i} \max(0, s_j - s_i + \lambda_{ij}), \tag{5}$$

where $\lambda_{ij}$ is the margin calculated as the difference in log probabilities between the answers predicted by the LLM using visual tokens of the $i$-th and $j$-th granularities:

$$\lambda_{ij} = \frac{j-i}{|T|} \sum_{t=1}^{T} (\log P(x_t|\mathbf{X}_v^i, \mathbf{X}_{instruct}, \mathbf{X}_{a,<t}) - \log P(x_t|\mathbf{X}_{\hat{v}}^j, \mathbf{X}_{instruct}, \mathbf{X}_{a,<t})). \tag{6}$$

When the preference of $\mathbf{X}_v^j$ is only slightly worse than $\mathbf{X}_v^i$, the margin will be small. Conversely, when $\mathbf{X}_v^j$ is significantly worse than $\mathbf{X}_v^i$, the margin will correspondingly increase. In this way, we can dynamically adjust the margin to obtain adaptively penalty degrees between different pairs.

In addition to aligning with the LMM preference ranking, it is also desirable for the router to select the optimal visual granularity. Therefore, we add a cross-entropy loss to let the router learn the prediction of granularity with the highest log probability from the LMM, defined as follows:

$$k = \arg\max_i \sum_{t=1}^{T} \log P(x_t|\mathbf{X}_v^i, \mathbf{X}_{instruct}, \mathbf{X}_{a,<t}), \tag{7}$$

$$\mathcal{L}_{ce} = -\log P(g_k|\overline{\mathbf{X}}_v, \overline{\mathbf{X}}_{instruct}). \tag{8}$$

In summary, the total loss is defined as the weighted sum of two losses:

$$\mathcal{L}_4 = \mathcal{L}_{rank} + \alpha\mathcal{L}_{ce}, \tag{9}$$

where $\alpha$ is the hyperparameters used to maintain the balance between the ranking loss $\mathcal{L}_{rank}$ and cross-entropy loss $\mathcal{L}_{ce}$.

## 4 EXPERIMENTS

### 4.1 SETTINGS

**Training Datasets** Note that in this work, we mainly focus on investigating the effectiveness of automatic visual granularity selection in reducing the number of visual tokens and improving model performance. Therefore, during the first stage, we also use CC-595K image-text pairs for model training, as implemented in LLaVA-NeXT (Liu et al., 2024b). In the subsequent training stages, we also hope to use the same data as LLaVA-NeXT. However, the real user interaction data used for visual instruction fine-tuning in LLaVA-NeXT are not open-sourced, so we opt to extract 200K samples from ALLaVA (Chen et al., 2024a) dataset as a substitute. Although LLaVA-NeXT replaces TextVQA (Singh et al., 2019) with DocVQA (Mathew et al., 2021) and SynDog-EN (Kim et al., 2022), the TextVQA has already been included in the training data of most existing LMMs. Consequently, we choose to retain it to ensure a fair comparison with other models.[3] In total, the visual instruction fine-tuning data we use contains 1M image-text pairs.

**Implementation Details** Following Liu et al. (2024b), we use CLIP ViT-L/14 as the visual encoder, Vicuna-7B (Chiang et al., 2023) as the LLM. We set the filtered instruction token number $k$ to 32 and the cross-entropy loss weight $\alpha$ to 0.1. The impact of these two hyperparameters on model performance is discussed in Section 4.6. In our first two stages, we adopt the same training settings as LLaVA-NeXT for pre-training and visual instruction fine-tuning. In the third stage, the learning rates for the visual encoder and other modules are set to $2\times10^{-5}$ and $1\times10^{-5}$, respectively, with a batch size of 128. In the fourth stage, the learning rate for the visual granularity router is set to $1\times10^{-3}$, with a batch size of 128. More details of the training process are provided in Appendix A.2.

---

[3]Our data recipe follows Open-LLaVA-NeXT (Lin & Long, 2024).

Table 1: Comparison with LMMs of the same size on general VQA benchmarks and text-oriented VQA benchmarks. The best results are marked in bold, and the second best results are underlined.

| Model | LLM | General VQA | | | Text-oriented VQA | | | |
| | | GQA | ScienceQA | VizWiz | TextVQA | ChartQA | DocVQA | AI2D |
| --- | --- | --- | --- | --- | --- | --- | --- | --- |
| *Standard-resolution LMMs* | | | | | | | | |
| InstructBLIP (Dai et al., 2023) | Vicuna-7B | 49.2 | 60.5 | 34.5 | - | - | - | - |
| IDEFICS-9B (Team, 2023) | LLaMA-7B | 38.4 | - | 35.5 | 25.9 | - | - | - |
| Qwen-VL (Bai et al., 2023) | Qwen-7B | 59.3 | 67.1 | 35.2 | 63.8 | 65.7 | 65.1 | 62.3 |
| Qwen-VL-Chat (Bai et al., 2023) | Qwen-7B | 57.5 | 68.2 | 38.9 | 61.6 | **66.3** | 62.6 | 57.7 |
| InternVL-Chat (Chen et al., 2023) | Vicuna-7B | 62.9 | - | 52.5 | 57.0 | - | - | - |
| mPLUG-Owl2 (Ye et al., 2023) | LLaMA2-7B | 56.1 | 68.7 | 54.5 | 58.2 | - | - | - |
| MQT-LLAVA (Hu et al., 2024) | Vicuna-7B | 61.6 | 67.6 | 53.1 | - | - | - | - |
| LLaVA-1.5 (Liu et al., 2024a) | Vicuna-7B | 62.0 | 66.8 | 50.0 | 58.2 | - | - | - |
| *High-resolution LMMs* | | | | | | | | |
| SPHINX-2k (Lin et al., 2023) | LLaMA2-7B | 63.1 | 70.6 | 44.9 | 61.2 | - | - | - |
| TextMonkey (Liu et al., 2024c) | Qwen-VL-7B | - | - | - | 65.9 | 58.2 | 64.3 | - |
| Mini-Gemini-HD (Li et al., 2024a) | Vicuna-7B | - | - | - | **68.4** | - | - | - |
| LLaVA-NeXT (Liu et al., 2024b) | Vicuna-7B | **64.2** | 70.1 | 57.6 | 64.9 | 54.8 | 74.4 | 66.6 |
| LLaVA-NeXT-$M^3$ (Cai et al., 2024) | Vicuna-7B | - | **72.5** | - | 63.1 | 59.0 | 72.6 | 66.7 |
| AVG-LLaVA | Vicuna-7B | 63.0 | 71.1 | **59.8** | 67.1 | **66.3** | **74.6** | **67.3** |

Table 2: Comparison with LMMs of the same size on general multimodal benchmarks. The best results are marked in bold, and the second best results are underlined.

| Model | LLM | MME | $MME^C$ | MMB | $MMB^{CN}$ | POPE | MMMU |
| --- | --- | --- | --- | --- | --- | --- | --- |
| *Standard-resolution LMMs* | | | | | | | |
| InstructBLIP (Dai et al., 2023) | Vicuna-7B | 1084.0 | 229.0 | - | - | - | 30.6 |
| Qwen-VL-Chat (Bai et al., 2023) | Qwen-7B | 1487.6 | 360.7 | 60.6 | - | - | - |
| InternVL-Chat (Chen et al., 2023) | Vicuna-7B | 1525.1 | - | - | - | 86.4 | - |
| mPLUG-Owl2 (Ye et al., 2023) | LLaMA2-7B | 1450.2 | - | 64.5 | - | - | - |
| MQT-LLAVA (Hu et al., 2024) | Vicuna-7B | 1434.5 | 353.6 | 64.3 | - | 84.4 | 34.8 |
| LLaVA-1.5 (Liu et al., 2023a) | Vicuna-7B | 1510.7 | - | 64.3 | 58.3 | 87.3 | - |
| *High-resolution LMMs* | | | | | | | |
| SPHINX-2k (Lin et al., 2023) | LLaMA2-7B | 1470.6 | 326.8 | 65.9 | - | 87.2 | - |
| OtterHD-8B (Li et al., 2023a) | Fuyu-8B | 1223.4 | 331.4 | 58.3 | - | 86.0 | - |
| Mini-Gemini-HD (Li et al., 2024a) | Vicuna-7B | 1546.0 | 319.0 | 65.8 | - | - | 36.8 |
| LLaVA-NeXT (Liu et al., 2024b) | Vicuna-7B | 1519.0 | 332.0 | 67.4 | 60.6 | 86.5 | 35.8 |
| LLaVA-NeXT-$M^3$ (Cai et al., 2024) | Vicuna-7B | - | - | 68.0 | - | 87.2 | 34.0 |
| AVG-LLaVA | Vicuna-7B | **1557.4** | **366.8** | **69.9** | **61.8** | **87.4** | **37.4** |

**Evaluations** We evaluate our model on three kinds of benchmarks: (1) **general VQA benchmarks**: GQA (Hudson & Manning, 2019), SciQA-Img (Lu et al., 2022), and VizWiz (Gurari et al., 2018); (2) **text-oriented VQA benchmarks**: TextVQA (Singh et al., 2019), ChartQA (Masry et al., 2022), DocVQA (Mathew et al., 2021), and AI2D (Kembhavi et al., 2016); and (3) **general multimodal benchmarks**: MME (Fu et al., 2023), MMB (Liu et al., 2023b), $MMB^{CN}$ (Liu et al., 2023b), POPE (Li et al., 2023c), and MMMU (Yue et al., 2023).

## 4.2 MAIN RESULTS

**General VQA Benchmarks** The results in Table 1 show that AVG-LLaVA outperforms all standard-resolution LMMs on the General VQA benchmarks and other high-resolution LMMs on VizWiz. Although it does not achieve the best results on GQA and ScienceQA, it is important to note that AVG-LLaVA uses fewer visual tokens compared to other high-resolution models, and this comparison will be detailed in Section 4.3.

**Text-oriented VQA Benchmarks** In this category of benchmarks, as illustrated in Table 1, except for TextVQA, AVG-LLaVA outperforms all other comparison models. Back to TextVQA, AVG-LLaVA achieves the second-best result, only trailing behind Mini-Gemini-HD. Notably, Mini-Gemini-HD utilizes more than twice the amount of data during the pretraining and approximately 1.5 times the amount of data during the visual instruction fine-tuning compared to AVG-LLaVA.

**General Multimodal Benchmarks** Compared to traditional VQA datasets, this type of benchmark covers a broader range of evaluation aspects, requiring models to possess more complex perception and reasoning capabilities. As summarized in Table 2, AVG-LLaVA surpasses all other models, exhibiting superior overall performance and highlighting its adaptability and effectiveness across various disciplines. Specifically, AVG-LLaVA outperforms the second best model by 9.4 and 6.1 on MME and $MME^C$, respectively, and by 1.9 and 1.2 on MMB and $MMB^{CN}$, respectively. Moreover, AVG-LLaVA's performance on the POPE and MMMU benchmarks demonstrates its ability to reduce hallucinations and perform complex reasoning.

## 4.3 COMPUTATIONAL EFFICIENCY

Table 3: Comparisons of AVG-LLaVA and LLaVA-NeXT in terms of the number of visual tokens and actual inference speed, both of which are tested on 8 V100 GPUs with a batch size of 1. AVG-LLaVA can reduce the number of visual tokens by up to 85.3% and is up to 2.53× faster than LLaVA-NeXT.

| Metric | General VQA | | | Text-oriented VQA | | | MLLM Benchmarks | | |
|---|---|---|---|---|---|---|---|---|---|
| | GQA | ScienceQA | VizWiz | TextVQA | ChartQA | AI2D | MME | MMB | MMMU |
| Token Per Grid ↓ | 80.0% | 26.4% | 54.9% | 92.3% | 99.1% | 14.7% | 69.3% | 30.0% | 29.9% |
| Speed ↑ | 1.14× | 1.77× | 1.41× | 1.04× | 0.97× | 2.53× | 1.19× | 1.87× | 1.79× |

To validate the effectiveness of dynamic visual granularity selection, we compare AVG-LLaVA with LLaVA-NeXT in terms of visual token number and inference speed across multiple benchmarks. Specifically, for each type of benchmark, we select three benchmarks for comparison, and report the reduction in the number of visual tokens per grid and the actual speedup during inference. As shown in Table 3, except for text-intensive VQA benchmarks that require very fine-grained visual information, such as TextVQA and ChartVQA, AVG-LLaVA significantly reduces the number of visual tokens and improves inference speed across other benchmarks. Particularly, on the AI2D benchmark, AVG-LLaVA achieves better performance than LLaVA-NeXT while using only 14.7% of the visual tokens, and the inference speed increased by 2.53 ×.[4] Notably, even with the addition of two extra modules, there is no significant slowdown in inference speed on the ChartVQA benchmark when using a comparable number of visual tokens. Moreover, AVG-LLaVA only increases the number of parameters by 1.66% compared to LLaVA-NeXT.

## 4.4 ROUTING VISUALIZATION

To further understand the differences in the granularity selection of AVG-LLaVA across different benchmarks, we visualize the proportion of visual tokens selected at each granularity level for all benchmarks. Figure 5 shows the visualization results, it is evident that different tasks tend to favor different visual granularity, which is consistent with our expectations. In the case of text-intensive benchmarks like TextVQA, ChartQA, and DocVQA, the model requires fine-grained visual information, so the router predominantly selects the finest visual granularity. On the other hand, for benchmarks involving object-level questions, such as AI2D and MMMU, the model may find it easier to answer correctly by utilizing coarse-grained visual information. Additionally, we observe that the granularities with 72 and 288 visual tokens are rarely selected. However, we find that incorporating these granularities of visual tokens helps the model progressively learn to utilize visual tokens with different granularity and allows it to more fully understand the differences between various visual granularities. Further verifications are detailed in Section 4.5.

## 4.5 ABLATION STUDY

In order to validate the effectiveness of our designed modules and training paradigm, we conduct the following ablation analysis.

**Adaptive Visual Granularity vs. Fixed Visual Granularity** We first delve into the proposed adaptive visual granularity router and report results in Table 4(a). It is clear that, compared to fixed

---

[4]We also present qualitative results in Appendix A.3 and illustrate the effectiveness of adaptive visual granularity.

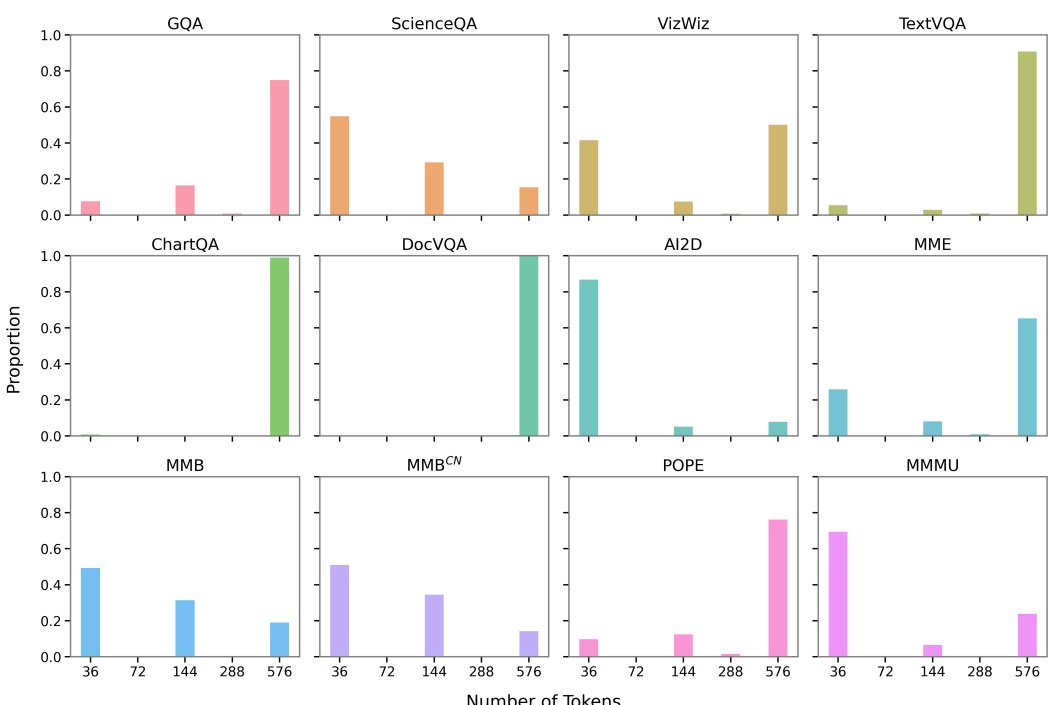

Figure 5: Visualization of the proportion for different granularity visual tokens.

Table 4: Ablation results on multiple benchmarks.

| Ablated Setting | | Ablated Details | Original Value | $\rightarrow$ | Changed Value | ScienceQA | ChartQA | MME | MMB |
|---|---|---|---|---|---|---|---|---|---|
| **AVG-LLaVA** | | | | | | 71.1 | 66.3 | 1557.4 | 69.9 |
| Architecture | (a) | Visual Granularity | Adaptive | | Fixed | 70.0 | 66.4 | 1554.5 | 68.7 |
| | (b) | Granularity Selection | Router | | Random | 69.7 | 56.8 | 1535.7 | 67.9 |
| | (c) | Router Input | Image + Instruction | | Image | 70.1 | 53.9 | 1525.2 | 69.0 |
| | (d) | Granularity Range | {36, 72, 144, 288, 576} | | {36, 144, 576} | 69.8 | 65.3 | 1547.7 | 66.3 |
| Training | (e) | Router Training | Feedback from LMM | | Visual Instruction Fine-tuning | 70.5 | 50.9 | 1514.8 | 68.6 |
| | (f) | Ranking Loss | ✓ | | ✗ | 70.1 | 64.8 | 1534.6 | 68.6 |
| | (g) | Cross-entropy Loss | ✓ | | ✗ | 70.2 | 66.3 | 1550.8 | 69.4 |

visual granularity, adaptive visual granularity shows significant improvement on ScienceQA, MME, and MMB. It is worth noting that, in addition to performance improvement, adaptive visual granularity can also significantly reduce the number of visual tokens and increase the model's inference speed, as reported in Section 4.3.

**Router Granularity Selection vs. Random Granularity Selection** In Table 4(b), we replace the granularity selected by the router with randomly-selected granularity. The results show that visual granularity router can indeed select a relatively appropriate granularity based on the input image and instruction, thereby significantly enhancing model performance.

**Impact of Router Input** The instruction plays a crucial role in granularity selection. To validate this, we remove the instruction from the router input. As shown in Table 4(c), a clear performance degradation rises when solely using image as input (e.g, -12.4 on ChartQA), illustrating the importance of choosing granularity based on input image and instruction.

**Impact of Granularity Range** In Section 4.4, we observe that granularities with 72 and 288 visual tokens are rarely selected, therefore we remove the visual tokens of these two granularities. As shown in Table 4(d), this change leads to a decrease in model performance, proving that introducing these granularities benefits the model's progressive learning to utilize features of different visual granularities and distinguish among various visual granularities.

**Impact of Router Training Methods** We directly train the router using visual instructions fine-tuning, applying the same loss function as in Stage 2. Unlike our original approach where the router is directly supervised by LMM feedback, this variant computes the loss on the LMM and backpropagates the gradient to the router using the Gumbel-Softmax technique (Jang et al., 2017). The results in Table 4(e) show that direct feedback from the LMM allows the router to better distinguish the advantages and disadvantages of different granularities, thereby enabling it to select an appropriate granularity.

**Importance of Ranking Granularity** In Table 4(f) and Table 4(g), we remove the cross-entropy loss and ranking loss during the fourth stage, respectively. The results indicate that both types of loss are beneficial to model training and are complementary to each other, between which the ranking loss is more crucial. This underscores the necessity to train the router by ranking granularity to align LMM feedback.

## 4.6 HYPERPARAMETER ANALYSIS

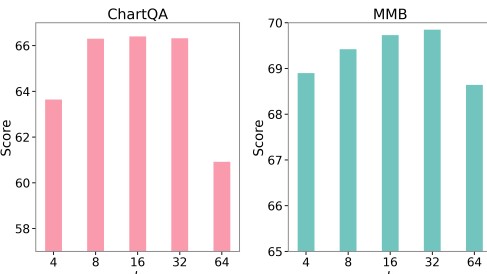
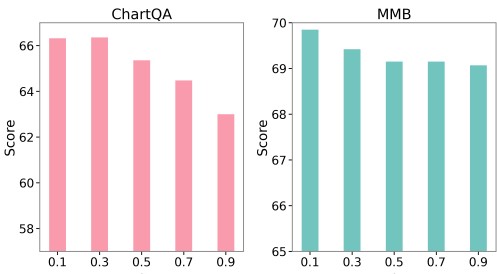

Figure 6: Influence of the filtered instruction token number $k$ on model performance, measured on ChartQA and MMB benchmarks.

Figure 7: Influence of the cross-entropy loss weight $\alpha$ on model performance, measured on ChartQA and MMB benchmarks.

We experimentally explore the influence of the filtered instruction token number $k$ and the cross-entropy loss weight $\alpha$ on model performance. As shown in Figure 6, the model performance is significantly affected when $k$ is too small or too large. This may be due to the fact that too few instruction tokens provide insufficient text information, while too many tokens will introduce more noise. Figure 7 indicates that our approach is relatively robust to $\alpha$ and setting a smaller $\alpha$ is able to consistently enhance model performance, making our training method easy to apply.

## 5 CONCLUSION

In this work, we propose AVG-LLaVA, an LMM that can adaptively select appropriate visual granularity based on input image and instruction. AVG-LLaVA builds upon LLaVA-NeXT by introducing a visual granularity scaler and a visual granularity router, which are used to obtain multi-granularity visual features and select the appropriate visual granularity based on image and instruction, respectively. Besides, we introduce RGLF, which aligns router-predicted probabilities of multiple granularities with LMM preferences by a ranking loss, effectively helping the model learn to distinguish between different granularities based on image and instruction. Experimental results show that AVG-LLaVA not only exhibits superior performance across 11 benchmarks, but also significantly reduce the number of visual tokens and speed up inference in tasks that do not require fine-grained information. Particularly, on the AI2D benchmark, it reduces the number of visual tokens by 85.3% and speeds the inference by 2.53×. We hope our work can inspire more attention to the visual granularity in LMMs.

While AVG-LLaVA has achieved good results, there is still considerable potential to be further explored. On text-intensive benchmarks, the model tends to select the finest-grained visual tokens, which may be due to the pooling directly reducing half of the tokens, resulting in significant differences in granularity size. In the future, we plan to design a more suitable granularity scaling network to provide richer visual granularities.

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

# A APPENDIX

## A.1 VISUAL GRANULARITY SELECTION ALGORITHM

In algorithm 1, we provide the detailed process of the router's granularity selection.

---

**Algorithm 1** Visual Granularity Selection Algorithm

---

**Require:** Multi-granularity visual tokens $\overline{\mathbf{X}}_v = [\mathbf{X}_v^1; \mathbf{X}_v^2; ...; \mathbf{X}_v^N]$, Instruction tokens $\mathbf{X}_{instruct}$,
   Visual tokens of original granularity $\mathbf{X}_v$
**Ensure:** Selected granularity visual tokens $\mathbf{X}_v^{selected}$.
 1: Obtain the filtered instruction tokens $\overline{\mathbf{X}}_{instruct} = \text{Top-}k(\text{cosine\_sim}(\mathbf{X}_{instruct}, \mathbf{X}_v))$
 2: Concatenate $\overline{\mathbf{X}}_v$ and $\overline{\mathbf{X}}_{instruct}$ to form the input for the router
 3: Apply a Transformer layer to facilitate token fusion $\mathbf{Z}_{fusion} = \text{Transformer}([\overline{\mathbf{X}}_v; \overline{\mathbf{X}}_{instruct}])$.
 4: Use an MLP to predict logits for each token $\mathbf{Z}_{out} = \text{MLP}(\mathbf{Z}_{fusion})$
 5: Aggregate the logits using a learnable weight matrix $\mathbf{Z}_{final} = \mathbf{W}\mathbf{Z}_{out}$.
 6: Compute the probability distribution using softmax $\mathbf{P} = \text{softmax}(\mathbf{Z}_{final})$.
 7: Identify the granularity with the highest probability $selected = \arg\max(\mathbf{P})$.
 8: **return** $\mathbf{X}_v^{selected}$

---

## A.2 TRAINING DETAILS

We list the training hyperparameters for all stages in Table 5.

| Hyperparameter | Stage 1 | Stage 2 | Stage 3 | Stage 4 |
|---|---|---|---|---|
| Data size | 595K | 1M | 1M | 1M |
| Batch size | 256 | 128 | 128 | 128 |
| lr | 1e-3 | 2e-5 | 1e-5 | 1e-3 |
| Vision encoder lr | - | 2e-6 | 2e-5 | - |
| lr schedule | | cosine decay | | |
| lr warmup ratio | | 0.03 | | |
| Weight decay | | 0 | | |
| Epoch | | 1 | | |
| Optimizer | | AdamW | | |
| DeepSpeed stage | 2 | 3 | 3 | 3 |
| Training cost (#Hours) | 4 | 17 | 65 | 14 |

Table 5: Hyperparameters of AVG-LLaVA.

## A.3 QUALITATIVE EVALUATION

As shown in Figures 8 and 9, we compare the visual granularity selected by the router and other visual granularities. In Figure 8, given the user-provided image and instruction, AVG-LLaVA selects the coarsest visual granularity through the router. It can be observed that compared to other granularities, the model's response with the coarsest granularity does not vary significantly. However, in Figure 9, with the given image and instruction, AVG-LLaVA selects the finest visual granularity. We find that coarser visual granularities could not generate a reasonably accurate poster description. These two examples demonstrate that AVG-LLaVA can adaptively select the appropriate visual granularity based on the image and instruction, thereby reducing the number of visual tokens, accelerating inference, and even improving model performance.

Figure 10 further shows several conversations between users and AVG-LLaVA.

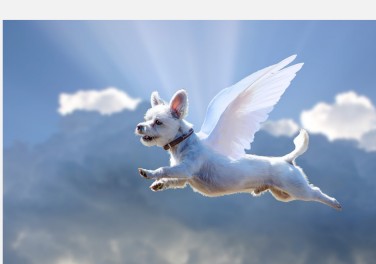

**User**: Why this image is unusual?

**AVG-LLaVA (36 selected by router)**: The image is unusual because it features a small white dog with wings, giving the appearance of a dog flying in the sky. This is not a natural occurrence, as dogs do not have the ability to fly. The image is likely a digitally manipulated or artistically created image, designed to evoke a sense of wonder and fantasy. The dog's wings and the fact that it is flying in the sky make the scene extraordinary and surreal, which is why it stands out as an unusual image.

**AVG-LLaVA (72)**: The image is unusual because it features a dog with wings, which is not a natural occurrence. Dogs are mammals and do not have the biological ability to fly like birds. The image is likely a digitally manipulated or artistically created image, designed to evoke a sense of wonder or fantasy. It is important to note that while the image is visually striking, it does not represent reality and should not be taken as an actual depiction of a dog with wings.

**AVG-LLaVA (144)**: The image is unusual because it features a small white dog with wings, which is not a natural occurrence. Dogs do not have the ability to fly, and it is highly unlikely to see a dog with wings in a real-life scenario. The image is likely a digitally manipulated or artistically created image, designed to evoke a sense of wonder, fantasy, or humor.

**AVG-LLaVA (288)**: The image is unusual because it features a dog with wings, which is not a natural occurrence. Dogs do not have the ability to fly like birds or other animals with wings. This creates a whimsical and fantastical scene that is not typically seen in reality. The dog's wings are likely added digitally or through some form of artistic manipulation, making the image stand out as an imaginative and creative representation.

**AVG-LLaVA (576)**: The image is unusual because it features a dog with wings, which is not a common sight. Dogs are typically known for their four legs and lack of wings, making this depiction of a dog with wings in the sky an imaginative and unexpected scene. The image could be a creative representation or an artistic interpretation, rather than a realistic portrayal of a dog.

Figure 8: Example of the finest visual granularity selected by the router compared to other visual granularities. The number in parentheses indicates the number of visual tokens. In this example, the responses from the model using both fine-grained and coarse-grained visual tokens show little difference, indicating that the model can adopt coarse visual granularity to significantly reduce the number of visual tokens and accelerate inference.

918
919
920
921
922
923
924
925
926
927
928
929
930
931
932
933
934
935
936
937
938
939
940
941
942
943
944
945
946
947
948
949
950
951
952
953
954
955
956
957
958
959
960
961
962
963
964
965
966
967
968
969
970
971

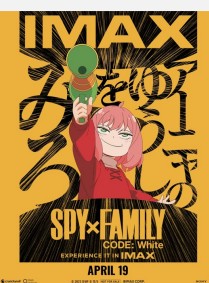

**User**: Please describe this poster in detail.

**AVG-LLaVA (36)**: The image is a promotional poster for the anime "Imax 90", which is written in large, bold letters at the top of the image. The title is followed by the phrase "IMAX 90", indicating the name of the anime and possibly the episode or series title. The background of the poster is a vibrant yellow, which contrasts with the black and white illustration of a character. This character is depicted with a determined expression, holding a sword in their right hand and a shield in their left, suggesting a warrior or combatant theme. The text and illustration are arranged in such a way that the title and character are the focal points, drawing the viewer's attention to the central theme of the anime.

**AVG-LLaVA (72)**: The image is a promotional poster for the anime "Maximilian", which is set to release on April 19th. The central figure in the poster is a young girl with pink hair, who is holding a megaphone. The text on the poster, written in Japanese, reads "IMAX MAXIMILIAN" and "4.19", indicating the title of the anime and its release date. The background of the poster is a vibrant yellow, adorned with black and white lines that add a dynamic visual element to the overall design. The girl's position and the megaphone suggest that she might be a character of significance in the anime, possibly a protagonist or a key figure. The use of bold colors and dynamic lines creates a sense of excitement and anticipation for the upcoming release.

**AVG-LLaVA (144)**: The image is a vibrant promotional poster for the anime "Impossible Spy Family". The central figure is a young girl with pink hair, who is holding a megaphone. She is dressed in a red jacket and a white shirt, adding a pop of color to the scene. The background is a striking yellow, adorned with black text that reads "IMAX" and "SPY FAMILY", indicating the title of the anime. The text is written in a bold, sans-serif font, making it stand out against the yellow backdrop. The overall design of the poster is simple yet eye-catching, with the girl's bright hair and the bold text creating a dynamic visual appeal.

**AVG-LLaVA (288)**: The image is a vibrant promotional poster for the anime "Maximum Spice!", which is set to premiere on April 19th. The central figure in the poster is a young girl with pink hair, who is holding a green microphone. She is dressed in a red jacket and a white shirt, adding a pop of color to the scene. The background is a bright yellow, adorned with Japanese text that reads "MAXIMUM SPICE!", "April 19th", and "スパイシーな家族", which translates to "Spicy Family". The text is written in a bold, black font, making it stand out against the yellow backdrop. The overall design of the poster suggests an exciting and spicy adventure is about to begin on April 19th.

**AVG-LLaVA (576 selected by router)**: The image is a promotional poster for the movie "Spy x Family". The central figure is a young girl with pink hair, who is holding a green megaphone. She is dressed in a red jacket and is smiling, giving off a cheerful vibe. The background is a vibrant yellow, adorned with Japanese text and a black and white striped pattern. The text on the poster reads "IMAX", "SPY x FAMILY", "CODE: WHITE", and "EXPERIENCE IT IN IMAX", indicating that the movie is being promoted as an IMAX experience. The poster also includes the release date "APRIL 19", suggesting when the movie will be released. The overall design of the poster is eye-catching and colorful, likely intended to attract the attention of potential viewers.

Figure 9: Example of the coarsest visual granularity selected by the router compared to other visual granularities. The number in parentheses indicates the number of visual tokens. In this example, when coarse-grained visual tokens are used, the model generates incorrect descriptions. This suggests that the model should select fine visual granularity for the image and instructions in order to achieve better accuracy.

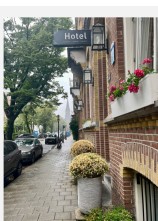

**User**: What is this place?

**AVG-LLaVA**: The place depicted in the image is likely a hotel, as indicated by the sign above the entrance. The architecture and the presence of the hotel sign suggest that this is a commercial area, possibly in a city or town center. The style of the building, with its brickwork and the design of the street, could imply that this location is in Europe, as such designs are common in many European cities. The overcast sky and the wet ground suggest that the photo was taken on a day with recent rainfall. The presence of the hotel, along with the parked cars and the residential-looking street, indicates that this area is a blend of commercial and possibly residential zones.

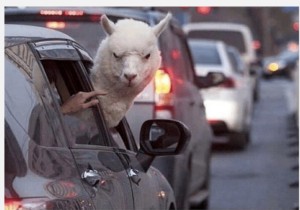

**User**: Why this image is funny?

**AVG-LLaVA**: The image is funny because it is an unexpected and absurd situation. Typically, one would not expect to see a llama, which is a large domesticated South American camelid, inside a car, especially sticking its head out of the window as if it were a human passenger. The juxtaposition of a llama behaving in a human-like manner in a typical urban setting, such as a traffic jam, is what makes the image humorous. It is a lighthearted and whimsical take on the unexpected and the anthropomorphism of animals, which often appeals to a sense of humor.

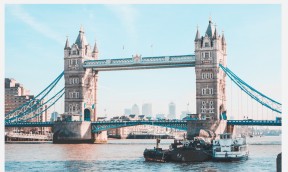

**User**: Tell me about this place.

**AVG-LLaVA**: The Tower Bridge is a combined bascule and suspension bridge in London, built between 1886 and 1894. It is a symbol of London and a popular tourist attraction. The bridge crosses the River Thames close to the Tower of London and has become an iconic symbol of London. The bridge is a bascule bridge, which means it can be raised to allow ships to pass. The suspension cables are anchored to the ground at either end of the bridge, and the towers are built on piers. The bridge is a testament to Victorian engineering and is still in operation today, although it is now primarily used for pedestrian traffic. The Tower Bridge is a testament to the engineering prowess of the Victorian era and has become a symbol of London's rich history and ongoing development.

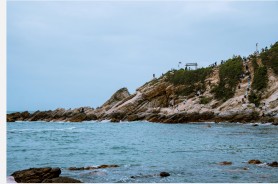

**User**: What are the things I should be cautious about when I visit here?

**AVG-LLaVA**: When visiting this rocky coastline with a large group of people, there are several things to be cautious about. Firstly, the rocky terrain can be slippery and uneven, so it's essential to be mindful of your footing and wear appropriate footwear to avoid accidents. Secondly, the presence of a large body of water nearby can pose risks such as strong currents, sudden changes in weather, or the possibility of falling into the water. It's crucial to be aware of the local weather conditions and any potential hazards in the water. Additionally, the large number of people in the area might lead to overcrowding, which can make it difficult to navigate and enjoy the scenery. It's important to respect the natural environment and other visitors by maintaining a safe distance and being mindful of your actions. Lastly, be aware of the potential for tides to change rapidly, which can affect the safety of the area and the people in it.

Figure 10: More Examples of conversations between users and AVG-LLaVA.

## A.4 ATTENTION MAP VISUALIZATION

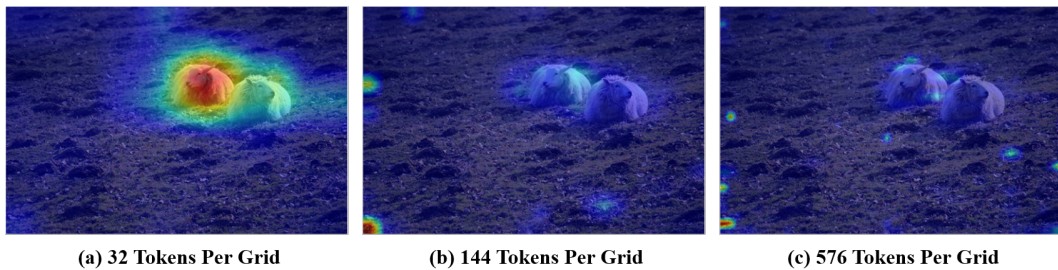

| (a) 32 Tokens Per Grid | (b) 144 Tokens Per Grid | (c) 576 Tokens Per Grid |

Figure 11: Attention maps of AVG-LLaVA at different visual granularities. The instruction is "How many sheep are there? Answer the question with a single word.". The number of visual tokens per grid selected by the router is 32.

To further understand how the appropriate granularity benefits the model in generating better answers, we visualize the attention map between the generated tokens and the visual tokens. The attention weights are calculated by accumulating the attention scores between image tokens and generated tokens across all layers and heads. As shown in Figure 11, when the instruction is "How many sheep are there? Answer the question with a single word," the attention weights for the visual granularity selected by the router are primarily focused on the two sheep, while the attention weights for other visual granularities are dispersed across the background. This means that selecting the appropriate visual granularity results in a clearer attention map with fewer noise points in the background area, indicating more precise focus on the relevant regions, thereby improving model performance.

## A.5 VISUALIZATION OF ROUTER SELECTION FOR DIFFERENT INSTRUCTIONS

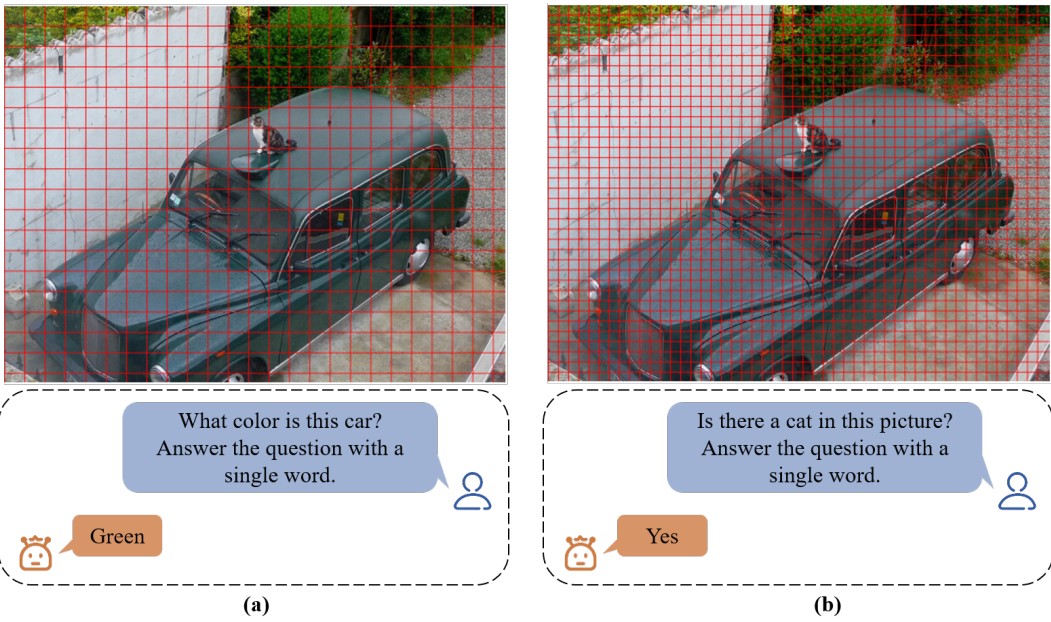

Figure 12: Visualization of granularity selection using different instructions.

As shown in Figure 12, we input the same image with different instructions and then visualize the selected visual granularity on the image, i.e., the number of patches. As can be seen, even for the same image, the router selects different visual granularities for different instructions. When asking

about the color of the car, the model does not require such fine-grained visual information, whereas when asking whether there is a cat, the model requires finer-grained visual information.

