# OpenReview forum: "AVG-LLaVA: A Large Multimodal Model with Adaptive Visual Granularity"
_ICLR.cc/2025/Conference — Submitted to ICLR 2025_

### Official Review · Reviewer_K4vy · 2024-10-29

**Soundness:** 3
**Presentation:** 3
**Contribution:** 2
**Rating:** 5
**Confidence:** 4

**Summary:**

The authors propose an adaptive visual granularity mechanism dubbed AVG-LLaVA.  Based on this assumption, they employ the visual granularity scaler to generate visual tokens with various granularities, and the visual granularity router to select the appropriate visual granularity. Besides, the paper introduces a training paradigm RGLF to enhance the router.  Comprehensive experiments are performed on various visual benchmarks to validate the effectiveness of the method.

**Strengths:**

1. The motivation is novel, and different prompts require information at different visual granularities. And the manuscript is explicit and well-organized.
2. The authors solve the problem of training the router in VLM directly and utilize the ranking loss to supervise, which is impressive.
3. Experimental validation is sufficient. The authors conduct comprehensive experiments on various tasks and show improvements, to validate the effectiveness of the method.

**Weaknesses:**

1. The method lacks novelty. (1) the multiple pooling operation in visual granularity scaler is very common, like the most classic SPPNet [1]. (2) the router operation has been proposed for many years.
2. Although the method sounds simple, the overall pipeline is complex. The stage 2 and 3 cost more training resources and time, where the vision encoder and LLM both are trained.

[1] He, K., Zhang, X., Ren, S. and Sun, J., 2015. Spatial pyramid pooling in deep convolutional networks for visual recognition. *IEEE transactions on pattern analysis and machine intelligence*, *37*(9), pp.1904-1916.

**Questions:**

1. Is it convenient to list their accuracy in Table 3 for further comparison? Besides, I want to know the absolute value of its actual speed.
2. I would like to see a visualization of actual token clipping, such as the image in Figure 1, and what the router results would be for different prompts.

---

> ### Author Response · Authors · 2024-11-20
> **Response to Reviewer K4vy**
>
> We thank you for your insightful feedback on improving the quality of our manuscript.
>
> **Response to W1**
>
> Although multiple pooling operations and router operations have appeared in other fields, their structures are not exactly the same as ours. Importantly, we are the first to propose an adaptive visual granularity selection method for LLMs.
> 1. Furthermore, our visual granularity scaler and visual granularity router are not the same as previous methods. For instance, the visual granularity scaler stacks 1x2 and 2x1 pooling operations, while the visual granularity router enables the information interaction between image and instruction tokens through a Transformer layer. Then, an MLP layer allows visual and instruction tokens to predict the granularity to be selected, and a Voter enables all tokens to vote for the selected granularity (whereas the router in traditional MOE only contains a linear layer).
> 2. Additionally, we propose a novel training paradigm, RGLF, which addresses the issue of poor performance due to the difficulty of distinguishing good and bad granularities during direct visual instruction fine-tuning. This aligns router probabilities of multiple granularities with LLM preferences.
> 3.  The ablation experiments on architecture and training in Table 4 also validate the effectiveness of AVG-LLaVA.
>
> **Response to W2**
>
> In Stage 2, we follow the setup of LLaVA-NeXT [1], training the visual encoder and LLM simultaneously, which are used by most current LMMs.
> We provide the training costs for each stage. We use a single node with 8 H800 GPUs (each with 80GB of memory) for training, and the costs are as follows:
>
> | Stage 1  | Stage 2   | Stage 3   | Stage 4   |
> |----------|-----------|-----------|-----------|
> | ~ 4 hour | ~ 17 hour | ~ 65 hour | ~ 14 hour |
>
> We have added this result in Table 5. Our computing resources are limited, and training will be faster with more resources in a multi-node, multi-GPU setup.
> Although the cost is increased compared to LLaVA-NeXT, these costs are justified because they significantly enhance model performance and reduce inference time without requiring additional large amounts of data. When a large number of users are accessing the model, the improvement in inference speed can save a lot of computing resources and bring higher benefits. This trade-off between increasing training costs and reducing inference costs is reasonable.
> In the future, we aim to explore methods for merging Stages 2, 3, and 4 to reduce training overhead. We also hope to explore LoRA training or methods for freezing certain modules during training.
>
> **Response to Q1**
>
> Thank you for your suggestion. We will list the performance changes of AVG-LLaVA compared to LLaVA-NeXT in Table 3 in the revised version, as follows:
>
> | Metric           | GQA   | ScienceQA | VizWiz | TextVQA | ChartQA | AI2D  | MME   | MMB   | MMMU  |
> |------------------|-------|-----------|--------|---------|---------|-------|-------|-------|-------|
> | Token Per Grid ↓ | 80.0% | 26.4%     | 54.9%  | 92.3%   | 99.1%   | 14.7% | 69.3% | 30.0% | 29.9% |
> | Speed ↑          | 1.14× | 1.77×     | 1.41×  | 1.04×   | 0.97×   | 2.53× | 1.19× | 1.87× | 1.79× |
> | Accuracy ↑       | -1.2  | +1.0      | +2.2   | +2.2    | +11.5   | +0.7  | +38.4 | +2.5  | +1.6  |
>
> We use the widely used LMMs evaluation tool, lmms-eval, for testing. The specific throughput inference speeds are as follows:
>
> | Model      | GQA           | ScienceQA      | VizWiz        | TextVQA       | ChartQA       | AI2D           | MME            | MMB           | MMMU          |
> |------------|---------------|----------------|---------------|---------------|---------------|----------------|----------------|---------------|---------------|
> | LLaVA-NeXT | 8.44 sample/s | 11.73 sample/s | 2.48 sample/s | 3.24 sample/s | 7.44 sample/s | 9.39 sample/s  | 8.60 sample/s  | 4.00 sample/s | 0.71 sample/s |
> | AVG-LLaVA  | 9.62 sample/s | 20.79 sample/s | 3.49 sample/s | 3.37 sample/s | 7.21 sample/s | 23.75 sample/s | 10.28 sample/s | 7.48 sample/s | 1.27 sample/s |
>
> **Response to Q2**
>
> Thank you for your feedback. We have added the visualization of visual granularity selected by the router under different instructions in Appendix A.5. As shown in Figure 12 of the paper (see page 20 of the newly submitted pdf), we input the same image with different instructions and then visualize the selected visual granularity on the image, i.e., the number of patches. As can be seen, even for the same image, the router selects different visual granularities for different instructions. For example, when asking about the color of the car, the model does not require such fine-grained visual information (router selects 144 visual tokens per grid), whereas when asking whether there is a cat, the model requires finer-grained visual information (router selects 576 visual tokens per grid).
>
> **Reference**
>
> [1] LLaVA-NeXT: What Else Influences Visual Instruction Tuning Beyond Data?

---

> > ### Comment · Reviewer_K4vy · 2024-11-23
> >
> > Thanks for the detailed rebuttal. Firstly, the difference between the visual granularity scaler and SPPNet is only the kernel size. Besides, I am still concerned that the training cost is larger compared to the baseline. Finally, thanks for your visualization.
> >
> > I will maintain my score.

---

> ### Author Response · Authors · 2024-11-23
> **Response to Reviewer K4vy**
>
> Thank you for your response. The main purpose of SPPNet is to handle input images of different sizes in image classification and object detection tasks. It utilizes max-pooling to merge features from images of varying sizes into a fixed number of visual features.
>
> In AVG-LLaVA, the visual granularity scaler is only a small part of the model and not the primary innovation of our work. SPPNet pools image features of different sizes into fixed sizes such as 4×4, 2×2, and 1×1. The only similarity between the visual granularity scaler and SPPNet lies in the use of multiple pooling operations. Beyond the visual granularity scaler, we introduced a **visual granularity router** consisting of a Transformer layer, an MLP layer, and a voter layer, which is designed to adaptively select the appropriate visual granularity based on the input image and instruction.
>
> In addition, we proposed **RGLF**, which addresses the challenge of poor performance due to difficulty in distinguishing different granularities during direct visual instruction fine-tuning. RGLF aligns the probabilities of multiple granularities in the router with the preferences of the LLM.
>
> **Overall, the similarity between AVG-LLaVA and SPPNet only lies in the use of multiple pooling operations in the visual granularity scaler module, which is a very small part of our work. To the best of our knowledge, our work is the first attempt to design an LMM that can adaptively select the appropriate visual granularity based on the input image and instruction.** Our primary contributions are the introduction of the visual granularity router and the novel RGLF training method, enabling an LMM to adaptively select the visual granularity based on the image and instruction. The results in Table 4 (a), (b), (c), (e), and (f) all demonstrate the effectiveness of the visual granularity router and RGLF.
>
> We acknowledge the increased training cost; however, training is conducted offline and only needs to be performed once. We believe this is a worthwhile trade-off, as a moderate increase in training cost can significantly improve inference speed.

---

### Official Review · Reviewer_UYRw · 2024-11-01

**Soundness:** 3
**Presentation:** 2
**Contribution:** 3
**Rating:** 6
**Confidence:** 5

**Summary:**

This work aims to enhance the LMM LLAVA-NeXT through improved visual granularity selection.
To achieve this, we introduce AVG-LLAVA, which consists of a visual granularity scaler,
a visual granularity router, and the RGLF training paradigm.
Experiments have been conducted to validate the effectiveness of the proposed method.

**Strengths:**

1. The research focus  is intriguing, particularly the aspects of visual granularity selection and the Ranking Granularity to Align LMM Feedback.

2. Experimental results demonstrate its effectiveness.

**Weaknesses:**

1. The training pipeline has become more complicated, moving from original two stages to four, which increases the training overhead despite the performance improvements.

2. I think the description of the main contributions is not well-articulated; it should better to include an algorithm, especially the Visual Granularity Router.

3. It would be beneficial to provide direct, rigorous evidence for the selection of granularity to illustrate the proposed method.

4. Providing visual examples that highlight the need for granularity, such as attention maps of visual tokens in the LLM, would be advantageous.

5. In Table 3, for ChartQA, the token per grid is 99.1%, while the speed is 0.97x without any increment.

**Questions:**

It should better to provide total token numbers of each method in main performance comparsion for each method.

---

> ### Author Response · Authors · 2024-11-20
> **Response to Reviewer UYRw (1/2)**
>
> We thank you for your insightful feedback on improving the quality of our manuscript.
>
> **Response to W1**
>
> We acknowledge the concerns about the additional computation costs.
> We provide the training costs for each stage. We use a single node with 8 H800 GPUs (each with 80GB of memory) for training, and the costs are as follows:
>
> | Stage 1  | Stage 2   | Stage 3   | Stage 4   |
> |----------|-----------|-----------|-----------|
> | ~ 4 hour | ~ 17 hour | ~ 65 hour | ~ 14 hour |
>
> We have added this result in Table 5. Our computing resources are limited, and training will be faster with more resources in a multi-node, multi-GPU setup.
>
> Although the cost is increased compared to LLaVA-NeXT, these costs are justified because they significantly enhance model performance and reduce inference time without requiring additional large amounts of data. When a large number of users are accessing the model, the improvement in inference speed can save a lot of computing resources and bring higher benefits. This trade-off between increasing training costs and reducing inference costs is reasonable. In the future, we also hope to explore methods for merging stages 2, 3, and 4 to reduce training overhead.
>
> **Response to W2**
>
> Thank you for your feedback. Following your suggestion, we have added the algorithm to Appendix A.1 in the newly submitted version of the paper.
>
> **Response to W3 and W4**
>
> Thank you for your constructive comments. We randomly sample 50 examples from each of the benchmarks: ScienceQA, ChartQA, MME, and MMB. Then, we conduct a manual review to determine whether the images needed to be carefully examined to answer the questions. Examples that need to be carefully examined require fine-grained visual information; otherwise, coarse-grained visual information is sufficient. The proportion of cases requiring careful image examination is as follows:
>
> | ScienceQA | ChartQA | MME | MMB |
> |-----------|---------|-----|-----|
> | 12%       | 92%     | 32% | 12% |
>
> Except for ChartQA, most of the other benchmarks only require a quick glance at the image to answer the questions. This indicates the potential to reduce the number of visual tokens. This observation aligns with the trend of granularity selection by the router in these benchmarks, as shown in Figure 5. Our experimental results also indicate that on coarse-grained benchmarks such as ScienceQA, MME, and MMB, using fewer visual tokens can lead to performance improvements.
>
> In addition, we have supplemented the experiments of visualizing the attention maps of model-generated tokens between visual tokens in Appendix A.4. We visualize the attention map between the generated tokens and the visual tokens. The attention weights are calculated by accumulating the attention scores between image tokens and generated tokens across all layers and heads. As shown in Figure 11 (see page 20 of the newly submitted pdf),  when the instruction is ``How many sheep are there? Answer the question with a single word,'' the attention weights for the visual granularity selected by the router primarily focus on the two sheep, while the attention weights for other visual granularities are dispersed across the background. **This means that selecting the appropriate visual granularity results in a clearer attention map with fewer noise points in the background area, indicating more precise focus on the relevant regions, thereby improving model performance.**
>
> In section A.3 of the Appendix, we provide a qualitative analysis to demonstrate the importance of granular selection. Besides, as shown in Table 4 (a), we compare the results of fixed visual granularity and adaptive granularity selection. It can be observed that adaptive granularity selection generally improves model performance. Additionally, as shown in Table 3, adaptive granularity selection can accelerate the model's inference speed.

---

> > ### Author Response · Authors · 2024-11-20
> > **Response to Reviewer UYRw (2/2)**
> >
> > **Response to W5**
> >
> > Due to the addition of a visual granularity scaler and a visual granularity router, there is some inference overhead. However, on the ChartQA benchmark, which requires fine-grained visual information, our inference speed only decreases by 3%, while on other benchmarks, we observe significant speed improvements. As mentioned in line 408 of the paper, the parameters of AVG-LLaVA increased by only 1.66%.
> >
> > **Response to Q1**
> >
> > Thank you for your feedback. Due to the fact that current high-resolution LMMs generally use dynamic image segmentation methods (such as the AnyRes technique), it is difficult for us to specify the total number of visual tokens for the comparison models. For example, when using high-resolution inputs, the number of visual tokens can reach up to 2880, while with low-resolution inputs, it can be as low as 576 tokens. For the most important comparison models, such as Mini-Gemini-HD, LLaVA-NeXT, and LLaVA-NeXT-M3, the maximum number of visual tokens is also 2880, and the number of tokens per grid is 576.
> > In summary, other high-resolution LMMs use 576$\times$$n$ visual tokens, while we use 576$\times$$n$$\times$$\alpha$ visual tokens, where $n$ is the number of sub-images and $\alpha$ is the token reduction ratio (for example, on AI2D, $\alpha$ is 14.7%).

---

### Official Review · Reviewer_je8e · 2024-11-02

**Soundness:** 3
**Presentation:** 3
**Contribution:** 2
**Rating:** 5
**Confidence:** 4

**Summary:**

This paper introduces a model that dynamically adjusts the granularity of visual tokens based on input images and instructions. This adaptive mechanism improves both efficiency and performance in multimodal tasks, reducing token usage and speeding up inference. The authors propose a novel training method, Ranking Granularity to Align LMM Feedback (RGLF), and test the model across 11 benchmarks. While the approach optimizes efficiency, concerns remain regarding scalability and performance trade-offs on certain tasks. The work offers promising advancements in multimodal learning.

**Strengths:**

The paper introduces a visual granularity scaler and router, which adaptively selects the appropriate granularity for visual tokens based on the input image and instructions. This adaptive selection mechanism is a significant advancement over static high-resolution LMMs, potentially improving both efficiency and accuracy in multimodal tasks.

**Weaknesses:**

1.	Lack of novelty: The motivation of this paper is highly similar to Matryoshka model, which also employs hierarchical token merging for visual token reduction, akin to token pruning in this paper. It seems that the difference is that the authors design an router to allocate weights to several granularities, which is incremental in terms of novelty.
2.	Insufficient experiments: This paper does not fully explore alternative approaches for granularity selection, such as task-specific fine-tuning or manual selection for certain tasks that might further improve performance.
3.	While the model's adaptive granularity selection is a strength, the architecture of the visual granularity router (involving multiple pooling layers, Transformer layers, and a voter layer) adds significant complexity and a substantial computational cost.
4.	The performance improvement is not superior across all benchmarks. For example, in GQA and ScienceQA, the proposed method underperforms slightly compared to some baselines, raising concerns about whether token reduction is always beneficial.
5.	Repeated Training Data: The training data for Stages 2, 3, and 4 are identical. Therefore, it is unclear whether the performance improvement is due to repeated training, akin to training for three epochs.
6.	Performance on OCR Tasks: As shown in Table 5, the visual tokens for OCR tasks are almost entirely retained, rendering the filter ineffective. The improvement in OCR tasks may primarily stem from repeated training.

**Questions:**

1.	The ablation study in Section 4.5 suggests a strong reliance on instruction tokens for granularity selection. Could the model's robustness be affected in situations where instructions are ambiguous or noisy? This is more important to the industry from my perspective.
2.	The benchmarks used are well-known public datasets. However, has the model been evaluated in real-world scenarios with less curated, noisier data? This would test its robustness in a more practical context.
3.	Training Cost: Provide details of the training costs associated with each of the four training stages.
4.	Comparative Experiment: Conduct a comparative experiment by training LLaVA-Next with repeated SFT data two or three times and present the detailed results.

---

> ### Author Response · Authors · 2024-11-20
> **Response to Reviewer je8e (1/3)**
>
> We thank you for your insightful feedback on improving the quality of our manuscript.
>
> **Response to W1**
>
> 1. Compared to the Matryoshka model, which requires **manually** setting the number of visual tokens, AVG-LLaVA can **adaptively** select the appropriate visual granularity based on the input image and instructions. This makes it more practical in real-world scenarios, as it is infeasible to experiment with every possible granularity due to the high cost. **Experimental results show that AVG-LLaVA surpasses the Matryoshka model in both performance and speed in most benchmarks.**
> 2. Furthermore, the hierarchical token merging method used in the Matryoshka model is not unique; for instance, it has been applied in the classic SPPNet [1]. In contrast to the Matryoshka model, we introduce a visual granularity scaler and a visual granularity router, designed specifically for granularity scaling and selection. The architecture also different, with the router consisting of a transformer layer, an MLP layer, and a voter layer, taking multi-granularity visual features and filtered instruction features as input. The primary goal is to enable adaptive visual granularity selection.
> 3. Additionally, we propose the RGLF training paradigm, addressing the challenge of poor performance in direct visual instruction fine-tuning, where the router struggles to distinguish between different granularities. This allows the router to better select the appropriate visual granularity based on the image and instructions.
> The ablation studies on architecture and training in Table 4 further demonstrate the effectiveness of AVG-LLaVA.
>
> **Response to W2**
>
> Thank you for the suggestion. We believe that exploring task-specific fine-tuning or manual selection could reduce the generality of the large multimodal model. These approaches may enhance performance for specific tasks but would require substantial effort for each new task, potentially undermining the model's adaptability and scalability across diverse applications. Moreover, manual selection is impractical in real-world applications, as it requires iterating through all granularities for each sample to select the optimal one, which would result in significant cost overhead. Our approach, focusing on adaptive granularity selection, is designed to maintain the model's flexibility and efficiency while ensuring robust performance across varied tasks.
>
> **Response to W3**
>
> As shown in Table 3, on the ChartQA benchmark, even though most of the visual tokens were retained, the model's speed only decreases by 3%. Moreover, as mentioned in line 408 of the paper, the parameters of AVG-LLaVA increase by only 1.66%. These observations indicate that the computational cost of the modules we introduced is minimal. On other benchmarks, AVG-LLaVA demonstrates significant speed improvements, especially on AI2D, where it achieves a 2.53x acceleration.
>
> **Response to W4**
>
> Although AVG-LLaVA shows a slight performance decrease compared to the best baselines on GQA and ScienceQA, it still achieves the third and second best results, respectively. Notably, as shown in Table 3, we reduce the number of visual tokens by 20% and 73.6% on GQA and ScienceQA, respectively, while accelerating by 1.14x and 1.77x.
> Furthermore, on the other 8 benchmarks, AVG-LLaVA outperformes all other baselines, demonstrating the generalizability of the method. The ablation experiments in Table 4 (a) also compare adaptive and fixed (576) approaches, showing that the adaptive approach outperforms the fixed one with fewer visual tokens.

---

> ### Author Response · Authors · 2024-11-20
> **Response to Reviewer je8e (2/3)**
>
> **Response to W5**
>
> Thank you for your constructive comments. We follow your suggestion and fine-tune the model three times in Stage 2. The experimental results on general VQA benchmarks and text-oriented VQA benchmarks are as follows:
>
> | Model             | GQA  | ScienceQA | VizWiz | TextVQA | ChartQA | DocVQA | AI2D |
> |-------------------|------|-----------|--------|---------|---------|--------|------|
> | LLaVA-NeXT        | 64.2 | 70.1      | 57.6   | 64.9    | 54.8    | 74.4   | 66.6 |
> | LLaVA-NeXT-3epoch | 64.6 | 69.9      | 58.3   | 63.9    | 66.3    | 75.1   | 65.3 |
> | AVG-LLaVA         | 63.0 | 71.1      | 59.8   | 67.1    | 66.3    | 74.6   | 67.3 |
>
> The experimental results on general multimodal benchmarks are as follows:
>
> | Model             | MME    | MME$^{C}$ | MMB  | MMB$^{CN}$ | POPE | MMMU |
> |-------------------|--------|-------|------|----------|------|------|
> | LLaVA-NeXT        | 1519.0 | 332.0 | 67.4 | 60.6     | 86.5 | 35.8 |
> | LLaVA-NeXT-3epoch | 1524.7 | 330.0 | 67.8 | 57.0     | 87.4 | 34.8 |
> | AVG-LLaVA         | 1557.4 | 366.8 | 69.9 | 61.8     | 87.4 | 37.4 |
>
> 1. It can be observed that although three repeated trainings result in improvements on 7 benchmarks (e.g., ChartQA and DocVQA), there is a considerable performance decline on 6 benchmarks (e.g., TextVQA and MMB$^{CN}$). This indicates that repeated training cannot improve the performance on all benchmarks.
> 2. AVG-LLaVA performs better than LLaVA-NeXT-3epoch on 9 benchmarks, is slightly worse on 2 benchmarks, and has a significant speed improvement, indicating that the advantage of AVG-LLaVA does not simply stem from repeated training.
>
> **Response to W6**
>
> As you mentioned, in OCR tasks, most of the visual tokens are retained, but we believe this is reasonable because such tasks generally require fine-grained visual information for text recognition. This indicates that the router is capable of distinguishing different inputs and selecting the most appropriate granularity, rather than favoring a single granularity.
> As mentioned in our response to W5, the performance of TextVQA declines after repeated training. This indicates that the performance improvement in OCR tasks is not solely due to repeated training but also benefits from multi-granularity instruction fine-tuning. Additionally, the results in Tables 3 and 4 (a) show that dynamic granularity selection significantly benefits both speed and performance on other tasks, demonstrating the generalizability of the method.
>
> **Response to Q1**
>
> It is reasonable for the model to require instructions for granularity selection, as shown in Figure 1. Even for the same image, different instructions may require different visual granularities. To test the model's robustness in granularity selection based on instructions, we applied random dropout and noise perturbations to the instruction tokens input to the router.
> The experimental results of applying dropout to the instruction tokens on ScienceQA are as follows:
>
> | Drop ratio | Accuracy | Speed |
> |------------|----------|-------|
> | 0%         | 71.1     | 1.77× |
> | 15%        | 70.6     | 1.73× |
> | 30%        | 70.7     | 1.66× |
> | 45%        | 70.7     | 1.65× |
> | 60%        | 70.6     | 1.55× |
> | 75%        | 70.6     | 1.39× |
> | 90%        | 70.2     | 1.27× |
>
> The experimental results of adding Gaussian noise to the instruction tokens on ScienceQA are as follows:
>
> | Std  | Accuracy | Speed |
> |------|----------|-------|
> | 0    | 71.1     | 1.77× |
> | 0.01 | 70.7     | 1.69× |
> | 0.02 | 70.5     | 1.45× |
> | 0.03 | 70.5     | 1.28× |
> | 0.04 | 70.5     | 1.27× |
> | 0.05 | 70.3     | 1.25× |
>
> The experimental results above indicate that our granularity selection process is relatively robust to the instructions.
>
> **Response to Q2**
>
> Thank you for your valuable feedback. We acknowledge the importance of evaluating the model in real-world scenarios with less curated and noisier data to test its robustness. However, at this time, we do not have access to such datasets, as this type of data may involve personal privacy and requires ethical consideration.

---

> > ### Author Response · Authors · 2024-11-20
> > **Response to Reviewer je8e (3/3)**
> >
> > **Response to Q3**
> >
> > Based on your suggestion, We provide the training costs for each stage. We use a single node with 8 H800 GPUs (each with 80GB of memory) for training, and the costs are as follows:
> >
> > | Stage 1  | Stage 2   | Stage 3   | Stage 4   |
> > |----------|-----------|-----------|-----------|
> > | ~ 4 hour | ~ 17 hour | ~ 65 hour | ~ 14 hour |
> >
> > We have added this result in Table 5. Our computing resources are limited, and training will be faster with more resources in a multi-node, multi-GPU setup.
> >
> > Although the cost is increased compared to LLaVA-NeXT, these costs are justified because they significantly enhance model performance and reduce inference time without requiring additional large amounts of data. When a large number of users are accessing the model, the improvement in inference speed can save a lot of computing resources and bring higher benefits. This trade-off between increasing training costs and reducing inference costs is reasonable.
> >
> > **Response to Q4**
> >
> > Please refer to Response to W5.
> >
> > **Reference**
> >
> > [1] Spatial pyramid pooling in deep convolutional networks for visual recognition.

---

### Official Review · Reviewer_icSi · 2024-11-05

**Soundness:** 3
**Presentation:** 3
**Contribution:** 2
**Rating:** 5
**Confidence:** 4

**Summary:**

The paper presents AVG-LLaVA, a large multimodal model capable of adaptively selecting the appropriate visual granularity based on input images and instructions, aiming to enhance model performance and reduce the number of visual tokens to expedite inference. AVG-LLaVA extends LLaVA-NeXT with the addition of a visual granularity scaler and a visual granularity router, along with a novel training paradigm called RGLF, which aligns the router's predicted probabilities of multiple granularities with the preferences of the LMM through a ranking loss.

**Strengths:**

1. The paper introduces a novel approach to handle high-res images by adaptively selecting the appropriate granularity based on the input image and instruction. Also, it conducted experiments to develop the appropriate tuning practice (the training state 3 & 4) to unlock the potential of the new paradigm.
2. On multiple benchmarks, AVG-LLaVA demonstrates its efficacy. It can achieve better results compared to LLaVA-NeXT while consumes much less computations.

**Weaknesses:**

1. The training paradigm is complex. It incorporates two additional training stages, each requires extensive computation costs. The additional training cost may hinder this approach from being widely adopted.
2. The framework is not thoroughly investigates and the ablation study is not sufficient (see Questions).

**Questions:**

1. It's well known than finetuning VLMs on instruction tuning corpora with multiple epochs will typically improve the performance on benchmarks. The authors need to prove that the improvement cannot be simply attributed to 3x tuning epochs (corresponding to stage 2 to 4).
2. Achieving better performance with fewer visual tokens is not a usual case. Would you please include more qualitative & quantitative examples & analysis and discuss under which circumstances the VLM can achieve this?
3. The AVG-LLaVA framework can be easily extended to perform patch-wise granularity selection (for example, select different granularity for different patches). Would that be helpful to save more visual tokens under text-rich scenarios (the current AVG-LLaVA did not save much visual tokens for TextVQA and ChartQA).
4. Recently, Qwen2-VL proposed to use native dynamic resolution visual encoders (no patchify) to generate visual embeddings. It would be beneficial to show that AVG-LLaVA also works for that kind of visual encoders.

---

> ### Author Response · Authors · 2024-11-20
> **Response to Reviewer icSi (1/2)**
>
> We thank you for your insightful feedback on improving the quality of our manuscript.
>
> **Response to W1**
>
> We acknowledge the concerns about the additional computation costs.
> We provide the training costs for each stage. We use a single node with 8 H800 GPUs (each with 80GB of memory) for training, and the costs are as follows:
>
> | Stage 1  | Stage 2   | Stage 3   | Stage 4   |
> |----------|-----------|-----------|-----------|
> | ~ 4 hour | ~ 17 hour | ~ 65 hour | ~ 14 hour |
>
> We have added this result in Table 5. Our computing resources are limited, and training will be faster with more resources in a multi-node, multi-GPU setup.
>
> Although the cost is increased compared to LLaVA-NeXT, these costs are justified because they significantly enhance model performance and reduce inference time without requiring additional large amounts of data. When a large number of users are accessing the model, the improvement in inference speed can save a lot of computing resources and bring higher benefits. This trade-off between increasing training costs and reducing inference costs is reasonable.
>
> **Response to W1**
>
> Please refer to the following response to the question.
>
> **Response to Q1**
>
> Thank you for your feedback. We follow your suggestion and fine-tune the model three times in Stage 2. The experimental results on general VQA benchmarks and text-oriented VQA benchmarks are as follows:
>
> | Model             | GQA  | ScienceQA | VizWiz | TextVQA | ChartQA | DocVQA | AI2D |
> |-------------------|------|-----------|--------|---------|---------|--------|------|
> | LLaVA-NeXT        | 64.2 | 70.1      | 57.6   | 64.9    | 54.8    | 74.4   | 66.6 |
> | LLaVA-NeXT-3epoch | 64.6 | 69.9      | 58.3   | 63.9    | 66.3    | 75.1   | 65.3 |
> | AVG-LLaVA         | 63.0 | 71.1      | 59.8   | 67.1    | 66.3    | 74.6   | 67.3 |
>
> The experimental results on general multimodal benchmarks are as follows:
>
> | Model             | MME    | MME$^{C}$ | MMB  | MMB$^{CN}$ | POPE | MMMU |
> |-------------------|--------|-------|------|----------|------|------|
> | LLaVA-NeXT        | 1519.0 | 332.0 | 67.4 | 60.6     | 86.5 | 35.8 |
> | LLaVA-NeXT-3epoch | 1524.7 | 330.0 | 67.8 | 57.0     | 87.4 | 34.8 |
> | AVG-LLaVA         | 1557.4 | 366.8 | 69.9 | 61.8     | 87.4 | 37.4 |
>
> 1. It can be observed that although three repeated trainings result in improvements on 7 benchmarks (e.g., ChartQA and DocVQA), there is a considerable performance decline on 6 benchmarks (e.g., TextVQA and MMB$^{CN}$). This indicates that repeated training cannot improve the performance on all benchmarks.
>
> 2. AVG-LLaVA performs better than LLaVA-NeXT-3epoch on 9 benchmarks, is slightly worse on 2 benchmarks, and has a significant speed improvement, indicating that the advantage of AVG-LLaVA does not simply stem from repeated training.
>
>
> **Response to Q2**
>
> We randomly sample 50 examples from each of the benchmarks: ScienceQA, ChartQA, MME, and MMB. Then, we conduct a manual review to determine whether the images needed to be carefully examined to answer the questions. Examples that need to be carefully examined require fine-grained visual information; otherwise, coarse-grained visual information is sufficient. The proportion of cases requiring careful image examination is as follows:
>
> | ScienceQA | ChartQA | MME | MMB |
> |-----------|---------|-----|-----|
> | 12%       | 92%     | 32% | 12% |
>
> Except for ChartQA, most of the other benchmarks only require a quick glance at the image to answer the questions. This indicates the potential to reduce the number of visual tokens. This observation aligns with the trend of granularity selection by the router in these benchmarks, as shown in Figure 5. Our experimental results also indicate that on coarse-grained benchmarks such as ScienceQA, MME, and MMB, using fewer visual tokens can lead to performance improvements.
>
> In addition, we have supplemented the experiments of visualizing the attention maps of model-generated tokens between visual tokens in Appendix A.4. We visualize the attention map between the generated tokens and the visual tokens. The attention weights are calculated by accumulating the attention scores between image tokens and generated tokens across all layers and heads. As shown in Figure 11 (see page 20 of the newly submitted pdf),  when the instruction is ``How many sheep are there? Answer the question with a single word,'' the attention weights for the visual granularity selected by the router primarily focus on the two sheep, while the attention weights for other visual granularities are dispersed across the background. **This means that selecting the appropriate visual granularity results in a clearer attention map with fewer noise points in the background area, indicating more precise focus on the relevant regions, thereby improving model performance.**
>
> The ablation experiments in Table 4 (a) also demonstrate the effectiveness of adaptive granularity.

---

> > ### Author Response · Authors · 2024-11-20
> > **Response to Reviewer icSi (2/2)**
> >
> > **Response to Q3**
> >
> > Thank you for the constructive comments.
> > 1. Theoretically, the AVG-LLaVA framework can indeed be applied to patch-wise granularity selection. However, this would disrupt the relative positional relationships when transforming 2D image features into a 1D sequence. Concretely, current LMMs predominantly adopt the anyres technique, where the features of the sub-image are arranged according to their original spatial positions. Each row of image features is appended with a special line-break token before being flattened into a 1D sequence. If different merging strategies are applied to different parts of an image, it may lead to difficulties when flattening it into a 1D sequence. For example, in an image with 16×16 patches, if the top-left 8×8 patches are merged (i.e., coarse granularity) while the other patches remain unchanged, determining which row the merged patch belongs to would significantly impact the positional relationships in the flattened sequence. It becomes even more complex when different levels of granularity merging occur in other areas as well.
> > 2. Additionally, patch-wise granularity selection might substantially increase the difficulty of the model's learning process.
> > However, we agree that such an approach could be more adaptive and meaningful, making it a promising direction for further exploration.
> >
> > **Response to Q4**
> >
> > Thank you for your suggestion.
> > 1. Since Qwen2-VL was released on September 18 and the ICLR submission deadline was October 1, Qwen2-VL qualifies as concurrent work.
> > 2. Theoretically, AVG-LLaVA is also applicable to Qwen2-VL. However, as Qwen2-VL's data is closed-source and its scale is substantial, it is challenging for us to train and reproduce it from scratch.
> > 3. Directly using Qwen2-VL for subsequent-stage training may not yield optimal results. We plan to explore this further in the future.

---

### Author Response · Authors · 2024-11-23
**Looking Forward to Further Engagement with Reviewers**

We deeply appreciate the reviewers' efforts and valuable feedback on our work. While we have not received responses during the rebuttal period, we remain eager to address any remaining concerns or questions and welcome further discussions.

Over the past few days, we have worked diligently to address the reviewers' concerns and questions through additional experiments and detailed explanations. Therefore, we kindly hope that these clarifications and additional experiments will be considered in reevaluating our work.

We would like to express our heartfelt gratitude to all the reviewers for their time, effort, and invaluable contributions.

Sincerely,

Authors of Paper #4331

---

### Comment · Area_Chair_DQo7 · 2024-11-24

Dear Reviewers,

This is a friendly reminder that the discussion period will end on Nov 26th (Anywhere on Earth). If you have not already, please take a careful look at the other reviews and author responses, and comment on whether your original rating stands. Thank you.

Best, AC

---

### Comment · Area_Chair_DQo7 · 2024-11-28

Dear reviewers,

This is a friendly reminder that the discussion period has been extended until December 2nd. If you haven’t yet, we kindly encourage you to review the authors' rebuttal and messages at your earliest convenience and confirm whether your comments have been adequately addressed.

We greatly appreciate your service to this process.

Best, AC

---

### Meta-Review · Area_Chair_DQo7 · 2024-12-19

**Metareview:**

This paper introduces AVG-LLaVA, a large multimodal model designed to adaptively determine the appropriate level of visual granularity based on the input image and instruction. It builds upon LLaVA-NeXT and incorporates a visual granularity scaler and a visual granularity router, which work together to extract multi-granularity visual features and select the optimal granularity for a given image and instruction.  The paper received scores of 5, 5, 5, 6.  Mentioned positives include good motivation, intriguing approach, and promising results.  Mentioned negatives include incremental novelty, complex training paradigm, minor performance improvements, and insufficient experiments and analyses.  Only one of the reviewers engaged in the rebuttal, but felt their concerns were not adequately addressed.  The AC carefully considered the paper, rebuttal, and author messages.  The rebuttal and author messages address some concerns, particularly regarding experiments and analyses, but challenges related to incremental novelty and complex training persist.  The AC agrees with the reviewers that the paper does not meet the bar for acceptance to ICLR.

**Additional Comments On Reviewer Discussion:**

The paper received scores of 5, 5, 5, 6.  Mentioned positives include good motivation, intriguing approach, and promising results.  Mentioned negatives include incremental novelty, complex training paradigm, minor performance improvements, and insufficient experiments and analyses.  Only one of the reviewers engaged in the rebuttal, but felt their concerns were not adequately addressed.  The AC carefully considered the paper, rebuttal, and author messages.  The rebuttal and author messages address some concerns, particularly regarding experiments and analyses, but challenges related to incremental novelty and complex training persist.  The AC agrees with the reviewers that the paper does not meet the bar for acceptance to ICLR.

---

### Decision · Program_Chairs · 2025-01-22

Reject